# Stability of blood lead levels in children with low-level lead absorption

**Michelle Del Rio** [1©¤]*, **Christina Rodriguez**[1‡], **Elizabeth Alvarado Navarro**[1‡], **Chandima Wekumbura**[2‡], **Madhubhashini B. Galkaduwa**[2,3‡], **Ganga M. Hettiarachchi**[2©], **Christina Sobin**[1©]

**1** Department of Public Health Sciences, The University of Texas at El Paso, El Paso, Texas, United States of America, **2** Department of Agronomy, Kansas State University, Manhattan, Kansas, United States of America, **3** Kansas Department of Agriculture Laboratory, Manhattan, Kansas, United States of America

© These authors contributed equally to this work.
¤ Current address: Department of Environmental and Occupational Health, Indiana University, Bloomington, Indiana, United States of America
‡ CR, EAN, CW, and MBG are contributed equally to this work.
* midelrio@iu.edu

**Data Availability Statement:** A minimal anonymized data set necessary to replicate study findings along with SPSS syntax for modeling longitudinal models can be found in the Supporting

## Abstract

Current child blood lead (Pb) screening guidelines assume that blood lead levels (BLLs) are relatively stable over time, and that only youngest children are vulnerable to the damaging effects of lower-range BLLs. This study aimed to test the stability of lower-range ($\leq$ 10 µg/dL) child BLLs over time, and whether lower-range BLLs diminished with age among children aged 6 months to 16 years living in a lower-income neighborhood with a density of pre-1986 housing and legacy contamination. Age, sex, family income, age of residence, and/or residence proximity to point sources of Pb, were tested as potential additional factors. Capillary blood samples from 193 children were analyzed by inductively coupled plasma mass spectrometry (ICPMS). Multiple imputation was used to simulate missing data for 3 blood tests for each child. Integrated Growth Curve models with Test Wave as a random effect were used to test BLL variability over time. Among N = 193 children tested, at Time 1 testing, 8.7% had the BLLs $\geq$ 5 µg/dL (CDC "elevated" BLL reference value at the time of data collection) and 16.8% had BLLs $\geq$ 3.5 µg/dL (2021 CDC "elevated" BLL reference value). Modeling with time as a random effect showed that the variability of BLLs were attributable to changes within children. Moreover, time was not a significant predictor of child BLLs over 18 months. A sex by age interaction suggested that BLLs diminished with age only among males. Of the additional environmental factors tested, only proximity to a major source of industrial or vehicle exhaust pollution predicted child BLL variability, and was associated with a small, but significant BLL increase (0.22 µg/dL). These findings suggest that one or two BLL tests for only infants or toddlers are insufficient for identifying children with Pb poisoning.

## Introduction

There is scientific and clinical consensus that no level of lead (Pb) exposure is "safe" for children [1]. Even lowest-level childhood Pb exposure is associated with irreversible adverse health

Information. For more additional information or questions contact Michelle Del Rio at midelrio@iu.edu.

**Funding:** Funding for this research came from the U.S. Department of Housing and Urban Development (HUD), Grant # TXLTS0010-18 (to CS & GH), the U.S. Environmental Protection Agency (EPA)(to CS). The study was also supported by funds from the University of Texas at El Paso (UTEP) Center for Environmental Resource Management, the UTEP Border Biomedical Research Center (BBRC)(to CS), the National Institutes of Health, National Institute for Child Health and Human Development (R21HD060120) (to CS), and the J. Edward and Helen M.C. Stern Professorship in Neuroscience (to CS), UTEP. The funders had no role in study design, data collection and analysis, decision to publish, or preparation of the manuscript. URLs of each funder website: U.S. Department of Housing and Urban Development: https://www.hud.gov/program_offices/healthy_homes/lbp/lts U.S. Environmental Protection Agency: https://www.epa.gov/ UTEP's Border Biomedical Research Center: https://www.utep.edu/science/bbrc/ UTEP's Center for Environmental Resource Management: https://www.utep.edu/cerm/ National Institutes of Health, National Institute for Child Health and Human Development: https://www.nichd.nih.gov/.

**Competing interests:** The authors have declared that no competing interests exist.

effects including poorer academic performance [2–5], lowered IQ [6, 7], stunted growth [8, 9], and increased behavioral problems [10–13]. Despite knowledge of the permanent health consequences of childhood Pb exposure, more than half of a million U.S. children continue to be exposed to dangerous lower-level Pb [14], and children living in lower-income and predominantly Black and/or Hispanic neighborhoods are disproportionately affected [15–20]. Based on frequencies of individual and community risk factors, it has been recently estimated that approximately 50% of U.S. children below the age of 6 have detectable levels of Pb in their blood [21]. Of note, the actual child BLL data needed to inform this estimate do not exist.

Before a public health problem can be solved, its incidence and prevalence must be correctly characterized. With regard to child Pb exposure, implicit assumptions that underlie current testing policies may be undermining our best efforts. In most states, screening guidelines test children once or twice at ages 1 and 2 years; in many fewer states, testing recommendations extend to children up to the age of 6 years. Despite multiple recalls, point-of-care devices are still widely used for one-time screening. When a child's BLL is determined to be below a locally designated level, Pb exposure risk is assumed to be low, and children may never be tested again. If children's BLLs in fact vary significantly over time, and do not decrease with age, these screening guidelines cannot be expected to adequately detect exposed children; from a public health perspective, our understanding of the scope of the problem is necessarily incomplete.

In a recent paper [22] we comprehensively reviewed current knowledge regarding the complexity of Pb absorption, transport, and disposition in children's bodies, how these would be expected to produce variable BLLs over time, and how these factors can continue to influence BLLs through adolescence. Moreover, these mechanisms differ broadly for inhaled as compared to ingested Pb. For example, when Pb is inhaled, transport from the lungs into the circulatory system depends on many shifting factors including Pb particulate size, length of exposure, frequency of exposure, age-based respiratory rate, where in the lungs Pb particulates are deposited, and importantly, individual differences in the development of the lungs and alveoli [23–25]. On the other hand, when Pb is ingested, markedly different chemical, biological, bio-physiochemical factors in the gut, and maturity of the GI tract, as well as behavioral factors related to dietary intake, dietary deficiencies, and hand-to-mouth and play habits, influence absorption and thus levels of Pb detected in blood samples [26–28]. Further complicating predictions, when Pb-contaminated household dust is inhaled as well as ingested–a plausible scenario–absorption mechanisms can be additive, antagonistic, and/or synergistic. Finally, in the bloodstream, it is well established that absorption into red blood cells is influenced by individual differences in genetic predisposition, specifically, common (ALAD and HPEPT2) genetic variants [18]; and by normal fluctuations in iron, calcium, and zinc levels [29]. Additional age dependent factors can facilitate or oppose the distribution of Pb into tissues, as well as re-release of Pb into the circulatory system.

In these ways, current biological evidence suggests the very high likelihood that children's BLLs should be assumed to vary over time. Very few if any studies, however, have focused attention on this question. This study aimed to test the stability of BLLs over time in a sample of children with BLLs < 10 μg/dL. Possible risk factors contributing to BLL variability were also examined including age, sex, living in a family with income at or below the U.S. poverty line, living in a home built before 1986; and living near point sources of Pb contamination. Given the extant biological evidence discussed above, we hypothesized that BLLs would vary significantly within children; that Pb exposure and Pb exposure risk would not diminish with age; and that older children would not differ from younger children in their vulnerability to Pb exposure.

## Materials and methods

### Ethics and consents

The data here presented were collected for a large research program that aimed to improve child Pb surveillance and home mitigation in high-risk neighborhoods in El Paso, TX. All methods, procedures, forms, and materials for the research program were reviewed and approved annually by The Institutional Review Board of the University of Texas, El Paso (# 1309985). Written informed consent to participate was obtained from parents prior to the study; written assent from each child was obtained immediately prior to BLL testing. All study forms and materials were available in Spanish and English versions. Researchers in this study were bilingual and throughout the study interacted with participants in their preferred language.

### Study design

This was an analytical, observational, prospective longitudinal study using a convenience sample of children ages 6 months to 16 years old. Data were collected in neighborhoods deemed at "high-risk" for child Pb exposure due to possible legacy contamination, lower-socioeconomic income level, and with a preponderance of homes built before 1986. (While the federal residential Pb-paint law was enacted in 1978, leaded pipes and Pb solder continued to be used until 1986.) The community was predominantly of Hispanic or Latinx descent (82.2%), with 55.5% of children living below the U.S. poverty threshold. Previous studies of children in these neighborhoods have shown relatively high proportions of children with elevated BLLs [18–20, 30]. As of 2020, 21 of 36 zip codes in this region continued to be areas targeted by the Texas Department of State Health Services for child Pb screening [31]. The research program was conducted between January 2018 and March 2020, ending prematurely due to COVID shutdowns.

### Recruitment, data collection, and analytics

Families were identified through parent outreach in two elementary schools from one independent school district, during community events, community health fairs, by door-to-door invitation, and by "word of mouth." First time and follow-up testing occurred during scheduled home visits. Families were compensated for their participation with a $20 gift card to a retail store upon completion of forms and child screenings. During the first home visit, parents completed a family demographics, home, and child characteristics form, including questions about the child's age, family composition, race/ethnicity, income, education, history of Pb testing, cognitive and/or behavioral diagnoses, home Pb paint and soil hazards, and child behaviors that could increase risk of Pb exposure.

Anthropomorphic measures (height, weight, waist and hip circumference, blood pressure) were measured prior to every blood sample draw, and BMI was calculated for children $\geq 2$ years of age. The blood sampling procedure followed a strict, multi-step protocol to ensure the collection of uncontaminated samples. Children's hands were first washed thoroughly with antibacterial soap and water and dried with paper towels, then thoroughly wiped with an industry-standard cloth designed to remove metals from the surface of the skin (D-Wipe ™, Esca Tech, Inc., Milwaukee, Wisconsin). Each finger-stick whole blood sample of 50-microliters (μL) was collected into a sterile EDTA (Safe-T-Fill®) or Heparinized (Safe-T-Fill®) capillary blood collection 125-milliliter tube. Samples were stored at 4 C˚ up to one week, until transported overnight to the laboratory for analysis. Data and blood samples were collected by specially trained undergrad and graduate-level students in environmental and health sciences

under the supervision of the program manager. The supervising program manager (MDR) had approximately 7 years of experience using this procedure with children (a pediatric phlebotomist is not required for finger stick blood collection according to CDC guidelines).

All blood Pb analyses were conducted by the Department of Agronomy at Kansas State University (Dr. Ganga Hettiarachchi, Ph.D., laboratory head) using inductively coupled plasma mass spectrometry (ICP-MS, Agilent 7500cx, Santa Clara, CA) [32]. Blood Pb results were recorded in micrograms per deciliter (µg/dL). Standards were prepared using stock solutions of Pb; Bi-209 was used as the internal standard element for Pb. Lyphochek Whole Blood Metals Control, Level 1 #527, Level 2 #528, and Level 3 # 529 (Bio-Rad Laboratories, Inc., Hercules, CA, U.S.A.) were used as the reference (QA/QC) samples to confirm percent recovery. Pb recoveries were in the range of 95–105%. The ICPMS limit of detection (LOD) for Pb was 0.04 µg/dL. BLL values below the ICP-MS detection limit of $\leq 0.04 \mu g/dL$ (9/393, 2.3%) were manually imputed using values from an online random number generator application that produced random numbers with two decimal places between the values of 0 and 0.05.

The original goal of the study was to test each child a minimum of 4 times every 3 to 4 months. The frequency of testing however was dependent on parents' schedules and testing was interrupted by COVID-19 pandemic community shutdowns in March 2020, resulting in BLL tests every 3 to 6 months, and up to 3 tests for some children (sample size details provided below).

## Study sample

From an initial recruited sample of 110 families with 223 children, 9 children did not provide assent at Time 1 BLL testing; 1 child was too young to participate (4.5 months); 2 children exceeded the enrollment age ($> 16$ years old); 3 children were not available during the scheduled test appointment; and 2 families could not be contacted after their initial enrollment ("lost to follow-up"). The sample that completed Time 1 baseline BLLs included 107 families and 206 children. Since this study focused on lower-range BLLs, children with initial BLLs $> 10$ µg/dL were excluded from analyses (13/206, 6.31% of tested children). The final BLL data sample included $N = 193$ children.

In order to improve consistency in time between tests, from 320 total BLL samples, 12 (0.04%) BLLs measured less than two months after a previous sample, were excluded. The final sample included $N = 193$ children with BLLs at Time 1; $N = 86$ children with BLLs at Time 2; and $N = 29$ children with BLLs at Time 3. This sample size was not large enough to meet basic statistical power expectations for longitudinal analyses, and multiple imputation (SPSS Multiple Imputation Function, Version 26, IBM, Armonk, NY) was used to complete missing values.

## Multiple imputation methods

Through analysis of missing patterns, the missing data were determined to be missing at random (MAR) and had non-monotone and monotone patterns of missingness. Fully conditional specification methods (FCS), set to generate random numbers using the Mersenne Twister method, and active generator initialization with a fixed value of 200000, were used [33]. For multiple imputation, at least 95% relative efficiency is preferred [34, 35]. For this study, 20 iterations provided a relative efficiency of 98% in all Individual Growth Curve (IGC) model analyses. Following previous recommendations [36, 37] continuous variables that were included as predictors in later models were centered before running multiple imputation. For comparison purposes, all longitudinal models were conducted with the original dataset of $N = 29$ children

with three time points, and with the multiple imputation dataset that yield three time points for $N = 193$ children.

## Data management and analysis

All data were entered and checked for accuracy and missing values in Microsoft Excel before importing to SPSS (macOS Version 26 IBM, Armonk, New York), for descriptive and comparative analyses. The analyses followed previous recommendations for generalized linear regression with random effects [38]; IGC modeling [39]; and guidelines in SPSS technical materials [40]. As expected for child BLL data, histograms and the skewness metric (2.18) showed that children's BLLs at Time 1 ($N = 193$) followed a positive skew. IGC modeling accommodated the non-normal distribution of the data and previous recommendations for IGC [39] suggested that it was an appropriate and efficient means for testing whether child BLLs varied significantly over time.

In this approach, analyses begin with an estimation of the amount of variability attributable to individual variation over time (only the significance of the intercept is tested). The amount of possible intraindividual variability is then estimated by calculating an interclass correlation coefficient and this estimate is compared to a criterium to assess whether an IGC model including TIME as a random effect is warranted. To examine possible additional predictors of BLL variability, 3 time-invariant predictor variables were defined and tested including living below the 2020 poverty line; living in a home built before 1986; and/or living near a polluting industry, interstate highway, or international bridge. All three predictor variables were binary variables and coded -1 = no, 1 = yes. For all statistical tests, an alpha level of .05 was used.

Sex and age were included as fixed effects in all models tested. The main analyses tested Level 1 and Level 2 models. The Level 1 model tested the amount of change in BLL within children (intraindividual differences) from all time points. Results were used to calculate an Intraclass Correlation Coefficient (ICC) to estimate the amount of variability attributable to individual differences; an ICC less than 0.25 signaled large intraindividual differences, in which case IGC models would not be likely to perform better than a traditional ANOVA [39]. The Level 2 (growth curve) model assessed significant variations in individual trajectory changes in BLL by time. This model tested whether child BLLs changed predictably (either increased or decreased) as a function of time. Level 1 and Level 2 models were calculated using both the original (3 test waves, $N = 29$) and imputed (3 test waves, $N = 193$) data sets. (Pooled estimates of fixed effects were given in SPSS for $N = 193$ dataset.) Non-significant Level 2 test wave models were not submitted to tests of higher-order change with additional predictors. The results of these first tests guided subsequent models examining whether additional non-time variant factors might predict child BLLS (poverty, older home, living near pollution source).

## Results

### Demographic and clinical characteristics

Demographic and clinical characteristics of the sample are shown in Table 1. There were approximately equal proportions of males and females, and their BLL distributions were similar. The sample predominantly self-identified as Hispanic/Latino/Mexican American.

Mean family size was approximately 5 members and median reported annual household incomes were between $20,000 and $25,000, below the U.S. median household income for 2020.

Table 2 shows family home characteristics. The median built year of homes was 1969 and ranged between 1900 and 2016. A majority of children (70.5%, 136/193) of children lived in a

**Table 1. Clinical and demographic characteristics of children at time 1, N = 193.**

| | Time 1 | | |
|---|---|---|---|
| | M (*n* = 91) (47.2%) | F (*n* = 102) (52.8%) | T (*n* = 193) |
| Age (SD) | 7.77 (± 4.29) | 8.19 (± 3.87) | 7.99 (± 4.07) |
| Weight[a] lbs. (SD) | 71.88 (± 48.70) (88/91) | 71.16 (± 40.19) (100/102) | 71.50 (± 44.26) (188/193) |
| Height[a] in. (SD) | 49.38 (±11.05) (87//91) | 50.32 (± 10.13) (101/102) | 49.88 (± 10.54) (188/193) |
| W/H Ratio[a] (SD) | 0.91(± 0.10) (86/91) | 0.88 (± 0.07) (100/102) | 0.90 (± 0.09) (186/193) |
| BMI[b] (SD) | 19.02 (± 7.17) (87/91) | 18.09 (± 4.53) (100/102) | 18.52 (± 5.91) (187/193) |
| SYS[b] (SD) | 108.48 (± 17.46) (83/91) | 108.20 (± 12.04) (99/102) | 108.33 (±14.72) (182/193) |
| DIA[c] (SD) | 61.59 (± 11.52) (83/91) | 64.77 (± 9.23) (99/102) | 63.32 (± 10.43) (182/193) |
| Pb BL (μg/dL) (SD) | 1.58 (± 1.45) (91/91) | 1.52 (± 1.47) (102/102) | 1.55 (± 1.45) (193/193) |
| Min. | 0.05 | 0.00 | 0.00 |
| Max. | 8.20 | 8.70 | 8.70 |
| 25 percentile | 0.80 | 0.50 | 0.70 |
| Median | 1.20 | 1.15 | 1.20 |
| 75 percentile | 1.90 | 1.90 | 1.90 |
| Children's Ethnicity | (91/91) | (102/102) | (193/193) |
| Hispanic | 47/91 (51.6%) | 46/102 (45.1%) | 93/193 (48.2%) |
| Latino | 9/91 (9.0%) | 10/102 (9.8%) | 19/193 (9.8%) |
| Mexican | 4/91 (4.4%) | 10/102 (9.8%) | 14/193 (7.3%) |
| Mexican American | 4/91 (4.4%) | 9/102 (8.8%) | 13/193 (6.7%) |
| White | 5/91 (5.5%) | 5/102 (4.9%) | 10/193 (5.2%) |
| African American | 2/91 (2.2%) | 0/102 (0.0%) | 2/193 (1.0%) |
| Black | 0/91 (0.0%) | 1/102 (1.0%) | 1/193 (0.5%) |
| Two or more ethnicities | 20/91 (22.0%) | 21/109 (20.6%) | 41/193 (21.2%) |
| Mother's Ethnicity | (91/91) | (102/102) | (193/193) |
| Hispanic | 46/91 (50.5%) | 42/102 (41.2%) | 88/193 (45.6%) |
| Latino | 9/91 (9.9%) | 9/102 (8.8%) | 18/193 (9.3%) |
| Mexican | 13/91 (14.3%) | 26/102 (25.5%) | 39/193 (20.2%) |
| Mexican American | 3/91 (3.3%) | 7/102 (6.9%) | 10/193(5.2%) |
| White | 8/91 (8.8%) | 8/102 (7.8%) | 16/193 (8.3%) |
| Black | 0/91 (0.0%) | 1/102 (1.0%) | 1/193 (0.5%) |
| Two or more ethnicities | 12/91 (13.2%) | 9/102 (8.8%) | 21/193 (10.9%) |
| Mother's Education Level | (91/91) | (102/102) | (193/193) |
| Less than High School Diploma | 23/91 (25.3%) | 32/102 (31.4%) | 55/193 (28.5%) |
| High School Diploma | 20/91 (22.0%) | 18/102 (17.6%) | 38/193 (19.7%) |
| Completed some college or technical training | 24/91 (26.4%) | 20/102 (19.6%) | 44/193 (22.8%) |
| Bachelor's Degree | 13/91 (14.3%) | 15/102 (14.7%) | 28/193 (14.5%) |
| More than Bachelor's Degree | 11/91 (12.1%) | 17/102 (16.7%) | 28/193 (14.5%) |
| Family Size (SD) | 5.12 (± 1.74) (91/91) | 4.91 (± 1.69) (102/102) | 5.01 (± 1.71) (193/193) |
| Household Annual Income (SD) | $40,854 (± 44,652) (73/91) | $45,122 (± 53,445) (80/102) | $43,287 (± $49,757) (153/193) |

[a] Children did not want to be measured or they had a physical disability that prevented proper measurement.

[b] Children were too young for BMI interpretation.

[c] Children did not stay still for accurate blood pressure readings.

**Table 2. Home characteristics of children, N = 193.**

| | M (*n* = 91) | F (*n* = 102) | Total |
|---|---|---|---|
| Home Built Year | | | |
| Min. | 1900 | 1900 | 1900 |
| Max. | 2016 | 2016 | 2016 |
| 25th percentile | 1937 | 1950 | 1947 |
| Median | 1962 | 1971 | 1969 |
| 75th percentile | 1998 | 1998 | 1998 |
| Home Built Before 1986 | 66/91 (72.5%) | 70/102 (68.6%) | 136/193 (70.5%) |
| Living Near Point Sources of Pb | 42/91 (46.2%) | 57/102 (55.9%) | 99/193 (51.3%) |
| Zip Code | | | |
| 79836 | 0/91 (0.0%) | 3/102 (2.9%) | 3/193 (1.6%) |
| 79849 | 4/91 (4.4%) | 1/102 (1.1%) | 5/193 (2.6%) |
| 79901 | 4/91 (4.4%) | 6/102 (5.9%) | 10/193 (5.2%) |
| 79902 | 20/91 (22.0%) | 22/102 (21.6%) | 42/193 (21.8%) |
| 79903 | 4/91 (4.4%) | 1/102 (1.1%) | 5/193 (2.6%) |
| 79904 | 1/91 (1.1%) | 1/102 (1.1%) | 2/193 (1.0%) |
| 79905 | 24/91 (26.4%) | 25/102 (24.5%) | 49/193 (25.4%) |
| 79907 | 6/91 (6.6%) | 11/102 (10.8%) | 17/193 (8.8%) |
| 79912 | 3/91 (3.3%) | 3/102 (2.9%) | 6/193 (3.1%) |
| 79915 | 3/91 (3.3%) | 2/102 (2.0%) | 5/193 (2.6%) |
| 79924 | 10/91 (11.0%) | 7/102 (6.9%) | 17/193 (8.8%) |
| 79925 | 1/91 (1.1%) | 4/102 (3.9%) | 5/193 (2.6%) |
| 79928 | 3/91 (3.3%) | 0/102 (0.0%) | 3/193 (1.6%) |
| 79930 | 1/91 (1.1%) | 2/102 (2.0%) | 3/193 (1.6%) |
| 79932 | 1/91 (1.1%) | 1/102 (1.1%) | 2/193 (1.0%) |
| 79934 | 1/91 (1.1%) | 0/102 (0.0%) | 1/193 (0.5%) |
| 79935 | 0/91 (0.0%) | 1/102 (1.1%) | 1/193 (0.5%) |
| 79936 | 3/91 (3.3%) | 5/102 (4.9%) | 8/193 (4.1%) |
| 79938 | 1/91 (1.1%) | 6/102 (5.9%) | 7/193 (3.6%) |
| 88063 | 1/91 (1.1%) | 1/102 (1.1%) | 2/193 (1.0%) |

home built before 1986 and 51.3% (99/193) lived near a polluting industrial site, heavily trafficked international bridge crossing, and/or major interstate highway. Of those living in older homes, 52.2% (71/136) also lived near a source of Pb contamination.

## Descriptive analyses of child BLLs

As shown in Table 3, at all time points, the rates of children with BLLs exceeding the two most recent CDC BLL reference values ($\leq$ 3.5 μg/dL, Oct. 2021 and $\leq$ 5 μg/dL, Jan 2012 –Sep 2021) were two- to five-fold higher than the 2.5% expected value [41].

For descriptive purposes, the change in BLLs between test points was calculated by dividing the absolute BLL difference of two time points by the initial compared BLL value, and then multiplying the result by 100. The percentages were then categorized into three categories including no change, less than 50% change, and equal to or greater than 50% change; proportions for each were calculated. Table 4 shows the results of these calculations. Among children who completed two BLL tests (*N* = 86), only 1.2% of children (1/86) had no change in BLLs between Time 1 and Time 2; BLL change in 26.7% (23/86) was less than 50%; among 72.1% (62/86), BLL change was 50% or greater. A Fisher's Exact test determined that there were no

**Table 3. Frequency of elevated BLLs for N = 193 dataset.**

| | Time 1 | Time 2 | Time 3 |
|---|---|---|---|
| | n = 193 | n = 86 | n = 29 |
| BLL Range | 0.00–8.70 | 0.30–9.80 | 0.02–6.30 |
| $\geq 3.5$ μg/dL[1] | 19 (9.8%) | 12 (20.0%) | 2 (6.9%) |
| $\geq 5$ μg/dL[2] | 9 (4.7%) | 6 (7.0%) | 2 (6.9%) |

[1] Current CDC BLL reference value, effective Oct. 2021

[2] Jan 2012—Sept 2021 CDC BLL reference value

differences in proportions of BLL change between males and females, $X^2$ (2, N = 86) = 1.73, p = .710 (75.0% vs 69.0% with BLL change of 50% or greater, for males and females respectively).

Among children with three test points (N = 29) the descriptive change estimates were similar. From Time 1 to Time 2, no children had zero change; for 37.9% (11/29) BLL change was less than 50%; for 62.1% (18/29) BLL change was 50% or more. Between Time 2 and Time 3, 6.9% (2/29) had no change; for 34.5% (10/29) BLL change was less than 50%; for 58.6% (17/29) BLL change was 50% or more. Between Time 1 and Time 3, no children had zero change; for 27.6% (8/29) BLL changed less than 50% change; for 72.4% (21/29) BLL change was 50% or more.

**Table 4. Descriptive comparison of BLL change for children with two time points (N = 86) and three time points (N = 29).**

| | Time 1 versus Time 2 (N = 86) | | |
|---|---|---|---|
| | Male | Female | Total |
| n | 44/44 | 42/42 | 86/86 |
| Mdiff | 1.55 | 1.59 | 1.57 |
| SDdiff | ± 1.40 | ± 1.63 | ± 1.51 |
| Min. diff | 0.10 | 0.00 | 0.00 |
| Max.diff | 5.60 | 8.20 | 8.20 |
| 25 percentile | 0.60 | 0.70 | 0.60 |
| Median | 1.00 | 1.10 | 1.05 |
| 75 percentile | 1.93 | 1.65 | 1.78 |
| % Change | | | |
| 0% | 0/44 (0.0%) | 1/42 (2.4%) | 1/86 (1.2%) |
| < 50% | 11/44 (25.0%) | 12/42 (28.6%) | 23/86 (26.7%) |
| ≥ 50% | 33/44 (75.0%) | 29/42 (69.0%) | 62/86 (72.1%) |

| | Time 1 versus Time 2 | | | Time 2 versus Time 3 | | | Time 1 versus Time 3 | | |
|---|---|---|---|---|---|---|---|---|---|
| | Male | Female | Total | Male | Female | Total | Male | Female | Total |
| n | 14/14 | 15/15 | 29/29 | 14/14 | 15/15 | 29/29 | 14/14 | 15/15 | 29/29 |
| Mdiff | 1.05 | 1.68 | 1.38 | 1.33 | 1.61 | 1.48 | 0.95 | 1.26 | 1.11 |
| SDdiff | ± 1.07 | ± 2.05 | ± 1.65 | ± 1.32 | ± 2.22 | ± 1.81 | ± 0.85 | ± 1.16 | ± 1.02 |
| Min. diff | 0.10 | 0.10 | 0.10 | 0.00 | 0.00 | 0.00 | 0.06 | 0.10 | 0.06 |
| Max. diff | 4.40 | 8.20 | 8.20 | 4.50 | 8.30 | 8.30 | 2.90 | 4.10 | 4.10 |
| 25 percentile | 0.40 | 0.60 | 0.55 | 0.38 | 0.50 | 0.45 | 0.20 | 0.40 | 0.35 |
| Median | 0.75 | 1.10 | 0.90 | 0.93 | 0.80 | 0.90 | 0.80 | 0.90 | 0.80 |
| 75 percentile | 1.33 | 1.30 | 1.30 | 1.86 | 1.70 | 1.74 | 1.49 | 1.87 | 1.59 |
| % Change | | | | | | | | | |
| 0% | 0/14 (0.0%) | 0/15 (0.0%) | 0/29 (0.0%) | 1/14 (7.1%) | 1/15 (6.7%) | 2/29 (6.9%) | 0/14 (0.0%) | 0/15 (0.0%) | 0/29 (0.0%) |
| < 50% | 6/14 (42.9%) | 5/15 (33.3%) | 11/29 (37.9%) | 5/14 (35.7%) | 5/15 (33.3%) | 10/29 (34.5%) | 5/14 (35.7%) | 3/15 (20.0%) | 8/29 (27.6%) |
| ≥ 50% | 8/14 (57.1%) | 10/15 (66.7%) | 18/29 (62.1%) | 8/14 (57.1%) | 9/15 (60.0%) | 17/29 (58.6%) | 9/14 (64.3%) | 12/15 (80.0%) | 21/29 (72.4%) |

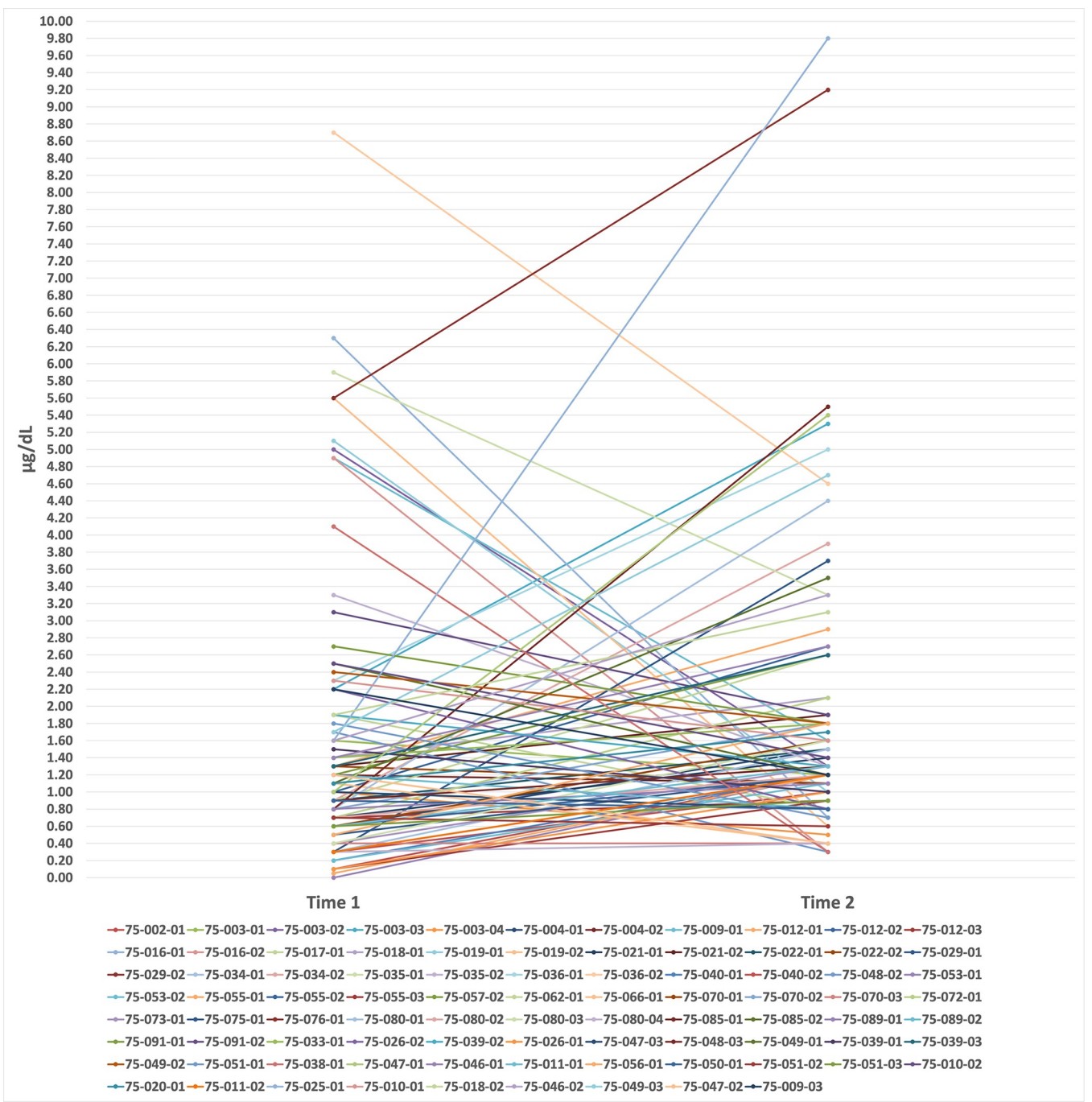

**Fig 1. BLLs overtime for children with two time points, N = 86.**

Again, Fisher's Exact Tests determined that proportion differences in BLL change between males and females were non-significant between Time 1/Time 2 and Time 2/Time 3 ($X^2$ (2, $N = 29$) = 2.207, $p = .332$), and between Time 1 and Time 3, $X^2$ (1, $N = 29$) = .901, $p = .343$. Linear graphs (Figs 1 and 2) show BLLs overtime for children with two time points ($N = 86$), and children with three time points ($N = 29$). As illustrated, a majority of BLLs changed between time points, but there was no clear direction of constant increases or decreases over time.

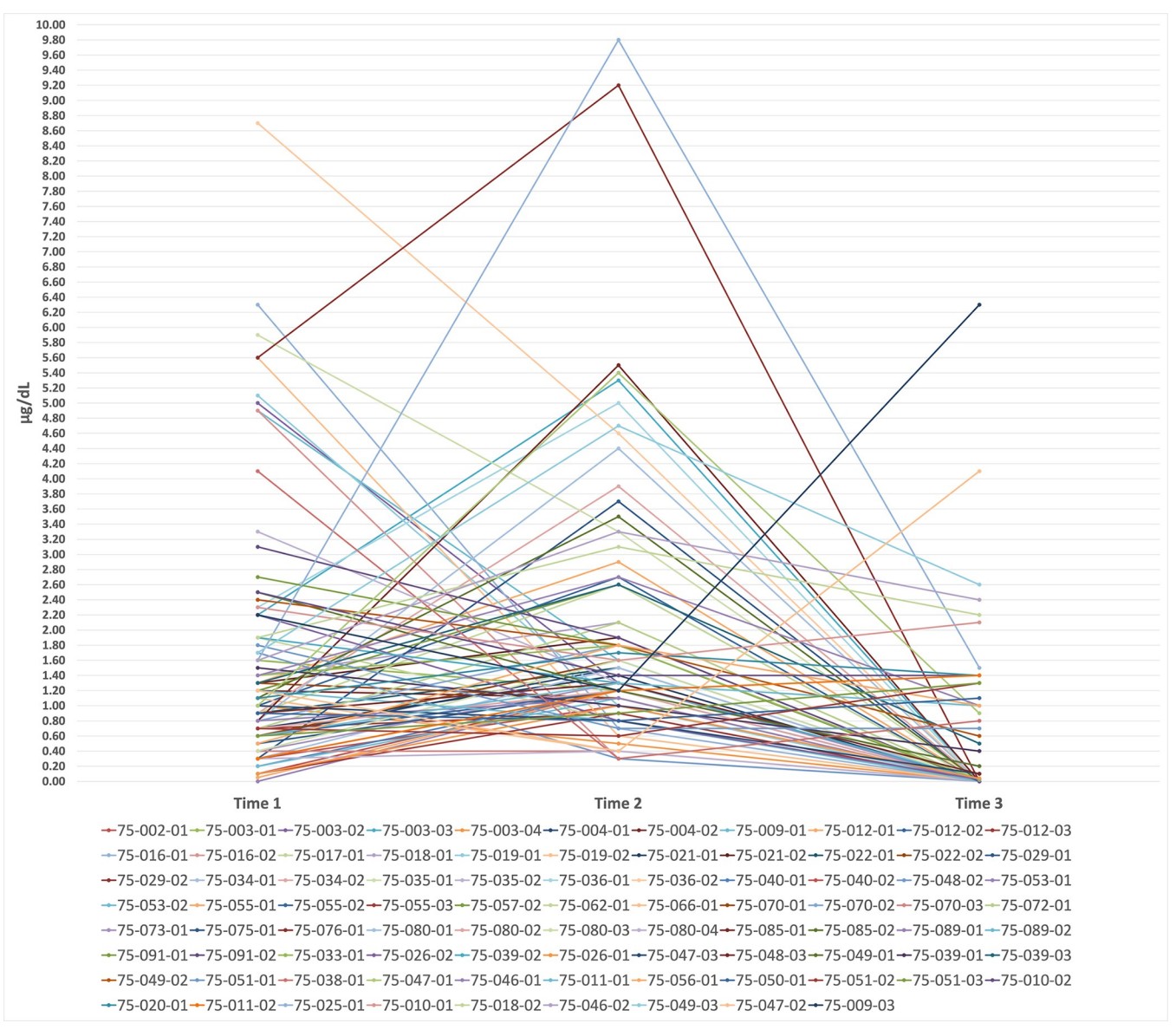

**Fig 2. BLLs overtime for children with three time points, N = 29.**

Older children (ages ≥ 7 years) did not differ from younger children (ages < 7 years) in their vulnerability to Pb exposure. A Pearson Chi-Square test in the original dataset determined that there were no differences in the rates of BLLs exceeding the most recent CDC BLL reference value (≤ 3.5 μg/dL) between younger and older children ($X^2$ (1, $n$ = 308 data points from all time points) = 0.047, $p$ = 0.828). Another Pearson Chi-Square test in the imputed dataset replicated the result ($X^2$, (1, n = 11888 data points from all time points) = 0.91, $p$ = 0.763).

## Longitudinal models

Level 1 model analysis was first calculated for the original dataset of children with three time points ($N$ = 29) (Table 5, row 1). The variability within children was statistically significant, $F$ (1, 29) = 95.69, $p$ < .001; the mean BLL was 1.57 μg/dL ($t$ = 9.78, $p$ < .001, 95% CI for the

**Table 5. Estimates of random (Time) and fixed effects (Sex and age) predicting child BLLs.**

| *Estimates of Fixed Effects* | | | | | 95% CI | | |
|---|---|---|---|---|---|---|---|
| | *Est.* | *SE* | *t* | *Sig.* | *Lower bound* | *Upper bound* | *RF%* |
| **Level 1 Model (*N* = 29)** | | | | | | | |
| *Intercept* | 1.57 | 0.16 | 9.78 | < .001 | 1.24 | 1.89 | - |
| **Level 1 Model (*N* = 193)** | | | | | | | |
| Intercept | 1.81 | 0.11 | 16.98 | < .001 | 1.60 | 2.02 | 0.98 |
| **Level 1 Model Random Effect TIME (*N* = 29)** | | | | | | | |
| Intercept | 1.80 | 0.39 | 4.60 | < .001 | 1.32 | 2.58 | - |
| TIME | -0.12 | 0.18 | -0.62 | .535 | 0.003 | 0.25 | - |
| **Level 1 Model Random Effect TIME (*N* = 193)** | | | | | | | |
| Intercept | 1.64 | 0.17 | 9.38 | < .001 | 1.29 | 1.98 | 0.99 |
| TIME | 0.087 | 0.10 | 0.85 | .400 | -0.12 | 0.29 | 0.98 |
| **Level 2 Model Fixed Effects SEX, AGE (*N* = 29)** | | | | | | | |
| Intercept | 1.50 | 0.12 | 12.22 | < .001 | 1.25 | 1.75 | - |
| AGE | 0.057 | 0.041 | 1.37 | .181 | -0.028 | 0.14 | - |
| SEX | 0.097 | 0.123 | 0.79 | .439 | -0.16 | 0.35 | - |
| SEX*AGE | -0.027 | 0.041 | -0.66 | .514 | -0.11 | 0.057 | - |
| **Level 2 Model Fixed Effects SEX, AGE (*N* = 193)** | | | | | | | |
| Intercept | 1.75 | 0.11 | 16.03 | < .001 | 1.53 | 1.97 | 0.98 |
| AGE | 0.006 | 0.023 | 0.282 | .778 | -0.034 | 0.051 | 0.99 |
| SEX | -0.12 | 0.08 | -0.14 | .890 | -0.18 | 0.15 | 0.99 |
| SEX *AGE | 0.056 | 0.023 | 2.59 | .010 | 0.014 | 0.10 | 0.99 |

*Note*. Significant p-value at alpha level .05 is highlighted in red.

difference was 1.24 to 1.89). The calculated Intraclass Correlation Coefficient was extremely low (0.03) and estimated that only 3% of all total variation in BLLs was due to between-child (interindividual) differences (97% of all total variation in BLLs was attributable to within child, "intra-individual" differences). These results indicated that an Individual Growth Curve for this sample was unlikely to perform better than a traditional ANOVA [39].

The Level 1 model was re-run using the imputed dataset of *N* = 193 and results are shown in Table 5, row 2. The results were similar. The imputed dataset Level 1 model showed that the variability within children was statistically significant, *p* < .001. On the other hand, the calculated Intraclass Correlation Coefficient increased to 0.32, as compared to the Level 1 model of the original dataset (0.03, *N* = 29). Thus, approximately 32% of all total variation in BLLs was due to between-child (interindividual) differences and approximately 68% of all total variation in BLLs was attributable to within child (intraindividual) differences. Thus, the Intraclass Correlation Coefficient using the imputed dataset met the suggested 0.25 threshold criteria [39] and a Level 2 Individual Growth Curve model with time as a random effect was calculated; the SUBJECTS option was used to model each subject's set of random parameters for the repeated effect of time, with an unstructured covariance matrix [33, 39, 42].

Table 5 summarizes estimates of fixed effects for the models predicting the influence of time, age, and sex, on child BLLs. The Level 2 models for *N* = 29 and *N* = 193 datasets were not significant predictors of child BLLs. The interaction of age and sex for *N* = 29 dataset was not significant (*Est.* = -0.027, *SE* = 0.041, *t* = -0.66, and *p* = .514); the model results for the imputed dataset showed a significant interaction of age and sex (*Est.* = 0.056, *SE* = 0.023, *t* = 2.59, *p* = .010). Since the age and sex interaction predicted child BLLs, these predictors were included in models with a third predictor.

**Table 6. Estimates of fixed effects for living below the poverty threshold, sex, and age, predicting child BLLs.**

| Estimates of Fixed Effects | | | | | 95% CI | | |
|---|---|---|---|---|---|---|---|
| | *Est.* | *SE* | *t* | *Sig.* | *Lower bound* | *Upper bound* | *RF%* |
| **Level 2 Model Fixed Effects AGE, SEX, POVERTY (*N* = 29)** | | | | | | | |
| Intercept | 1.27 | 0.29 | 4.42 | < .001 | 0.68 | 1.85 | - |
| AGE | 0.065 | 0.091 | 0.72 | .478 | -0.12 | 0.25 | - |
| SEX | 0.31 | 0.29 | 1.089 | .285 | -0.27 | 0.90 | - |
| POVERTY | 0.35 | 0.32 | 1.10 | .282 | -0.30 | 1.00 | - |
| AGE*SEX | 0.006 | 0.091 | 0.068 | .946 | -0.18 | 0.19 | - |
| AGE*POVERTY | -0.035 | 0.11 | -0.33 | .741 | -0.25 | 0.18 | - |
| SEX*POVERTY | -0.03 | 0.32 | -0.92 | .366 | -0.94 | 0.36 | - |
| AGE*SEX*POVERTY | -0.011 | 0.11 | -0.11 | .917 | -0.23 | 0.21 | - |
| **Level 2 Model Fixed Effects AGE, SEX, POVERTY (*N* = 193)** | | | | | | | |
| Intercept | 1.76 | 0.12 | 14.89 | < .001 | 1.52 | 1.99 | 0.98 |
| AGE | 0.011 | 0.025 | 0.42 | .675 | -0.039 | 0.06 | 0.98 |
| SEX | 0.023 | 0.093 | 0.31 | .757 | -0.15 | 0.21 | 0.99 |
| POVERTY | 0.034 | 0.10 | 0.34 | .735 | -0.16 | 0.24 | 0.99 |
| AGE*SEX | 0.065 | 0.023 | 2.79 | .006 | 0.019 | 0.11 | 0.99 |
| AGE*POVERTY | -0.022 | 0.024 | -0.94 | .349 | -0.069 | 0.024 | 0.99 |
| SEX*POVERTY | -0.05 | 0.10 | -0.5 | .619 | -0.24 | 0.14 | 0.99 |
| AGE*SEX*POVERTY | -0.022 | 0.024 | -0.91 | .362 | -0.07 | 0.026 | 0.99 |

*Note*. Significant p-value at alpha level .05 is highlighted in red.

Subsequent individual models tested three additional potential predictors of child BLLs (Tables 6–8) controlling for age and sex. These included living below the poverty line, living in a pre-1986 home, and living near a known source of Pb emissions. In the first 2 models, only the age and sex interaction was significant (and only in the imputed dataset). The final model showed a small but statistically significant effect on child BLLs of living near an environmental source of Pb pollution (*N* = 193, *Est.* = 0.22, *SE* = 0.093, *t* = 2.41, *p* = .026). The estimate suggested that living near a Pb point source increased child BLLs by an estimated 0.22 μg/dL (95% CI = 0.04 to 0.41).

## Discussion

Child Pb exposure is regarded by many as the longest-standing public health epidemic in U.S. history, and children living in lower-income settings are at a disproportionate risk of exposure and its many long-term health effects. We lack valid estimates of the true incidence and prevalence of the problem, and this undermines our capacity to initiate and sustain efforts that effectively manage the problem. It has been recently estimated that based on the frequencies of established risk factors, at any given time, over 50% of U.S. children are positive for Pb by current standards. We noted that the rates of children in this sample with elevated BLLs were 2- to 5-fold greater than the expected 2.5%. These and other accumulating results should be a clarion call to action.

Current testing practices and their underlying assumptions may be one factor contributing to the ongoing problem. Acceptance of one or two negative BLL tests at some point during early childhood is insufficient to assume that a child is at low or no risk of future Pb exposure. This study tested the implicit logical assumption of these policies, that child BLLs are relatively stable over time, and that child BLLs across the school-age years are not likely to change over

**Table 7. Estimates of fixed effects for living in a pre-1986 home, sex, and age, predicting child BLLs.**

| Estimates of Fixed Effects | | | | | 95% CI | | |
|---|---|---|---|---|---|---|---|
| | Est. | SE | t | Sig. | Lower bound | Upper bound | RF% |
| **Level 2 Model Fixed Effects AGE, SEX, OLDER HOME (N = 29)** | | | | | | | |
| Intercept | 1.57 | 0.13 | 11.66 | < .001 | 1.29 | 1.84 | - |
| AGE | 0.040 | 0.049 | 0.83 | .412 | -0.058 | 0.14 | - |
| SEX | 0.079 | 0.13 | 0.59 | .563 | -0.20 | 0.35 | - |
| OLDER_HOME | -0.12 | 0.13 | -0.92 | .367 | -0.40 | 0.15 | - |
| AGE*SEX | 0.022 | 0.049 | 0.47 | .641 | -0.076 | 0.12 | - |
| AGE*OLDER_HOME | 0.009 | 0.049 | 0.21 | .839 | -0.89 | 0.11 | - |
| SEX*OLDER_HOME | 0.074 | 0.13 | 0.55 | .587 | -0.2 | 0.35 | - |
| AGE*SEX*OLDER_HOME | -0.084 | 0.048 | -1.74 | .091 | -0.18 | 0.014 | - |
| **Level 2 Model Fixed Effects AGE, SEX, OLDER HOME (N = 193)** | | | | | | | |
| Intercept | 1.74 | 0.12 | 14.92 | < .001 | 1.51 | 1.97 | 0.08 |
| AGE | 0.012 | 0.027 | 0.46 | .644 | -0.04 | 0.065 | 0.98 |
| SEX | 0.017 | 0.093 | 0.19 | .853 | -0.17 | 0.20 | 0.99 |
| OLDER_HOME | 0.030 | 0.097 | 0.30 | .761 | -0.16 | 0.22 | 0.99 |
| AGE*SEX | 0.053 | 0.024 | 2.19 | .029 | 0.006 | 0.10 | 0.99 |
| AGE*OLDER_HOME | -0.011 | 0.024 | -0.46 | .645 | -0.059 | 0.036 | 0.99 |
| SEX*OLDER_HOME | -0.07 | 0.093 | -0.76 | .447 | -0.25 | 0.11 | 0.99 |
| AGE*SEX*OLDER_HOME | 0.008 | 0.025 | 0.31 | .754 | -0.042 | 0.057 | 0.99 |

*Note.* Significant p-value at alpha level .05 is highlighted in red.

**Table 8. Estimates of fixed effects for living near a known pb emissions site, sex, and age, predicting child BLLs.**

| Estimates of Fixed Effects | | | | | 95% CI | | |
|---|---|---|---|---|---|---|---|
| | Est. | SE | t | Sig. | Lower bound | Upper bound | RF% |
| **Level 2 Model Fixed Effects AGE, SEX, INDUSTRY (N = 29)** | | | | | | | |
| Intercept | 1.50 | 0.12 | 12.12 | < .001 | 1.25 | 1.76 | - |
| AGE | 0.059 | 0.04 | 1.38 | .178 | -0.03 | 0.15 | - |
| SEX | 0.10 | 0.12 | 0.81 | .425 | -0.15 | 0.35 | - |
| INDUSTRY | 0.16 | 0.12 | 1.31 | .200 | -0.09 | 0.42 | - |
| AGE*SEX | -0.040 | 0.04 | -0.94 | .353 | -0.12 | 0.047 | - |
| AGE*INDUSTRY | -0.024 | 0.04 | -0.56 | .583 | -0.011 | 0.064 | - |
| SEX*INDUSTRY | 0.079 | 0.12 | 0.63 | .532 | -0.18 | 0.33 | - |
| AGE*SEX*INDUSTRY | -0.021 | 0.04 | -0.49 | .629 | -0.11 | 0.066 | - |
| **Level 2 Model Fixed Effects AGE, SEX, INDUSTRY (N = 193)** | | | | | | | |
| Intercept | 1.68 | 0.11 | 15.49 | < .001 | 1.47 | 1.90 | 0.98 |
| AGE | 0.005 | 0.023 | 0.19 | .847 | -0.041 | 0.050 | 0.98 |
| SEX | 0.008 | 0.085 | 0.092 | .927 | -0.16 | 0.17 | 0.99 |
| INDUSTRY | 0.22 | 0.093 | 2.41 | .017 | 0.041 | 0.41 | 0.99 |
| AGE*SEX | 0.048 | 0.021 | 2.23 | .026 | 0.006 | 0.090 | 0.99 |
| AGE*INDUSTRY | -0.017 | 0.021 | -0.78 | .435 | -0.058 | 0.025 | 0.99 |
| SEX*INDUSTRY | -0.045 | 0.087 | -0.058 | .602 | -0.21 | 0.12 | 0.99 |
| AGE*SEX*INDUSTRY | 0.028 | 0.022 | 1.28 | .202 | -0.015 | 0.070 | 0.99 |

*Note.* Significant p-value at alpha level .05 is highlighted in red.

time. This study focused on children with lower-level Pb absorption (BLLs ≤ 10 μg/dL) living in high-risk neighborhoods.

Longitudinal data collection for this study was interrupted by the COVID pandemic and all analyses were conducted with all of the available collected data, and a complete imputed dataset with 3 data points for each of 193 children. Simple descriptive analyses and IGC models with time as a random effect and controlling for sex and age, showed that children's BLLs varied significantly and randomly. Supporting the results of the longitudinal models, we noted that a majority of children had equal to or greater than a 50% change in BLL at 3 time points. Moreover, BLLs across this wide age range (6 months to 16 years) did not increase or decrease with time, suggesting that children did not become less vulnerable to Pb exposure as they aged.

IGC models with time as a repeated measure were also calculated to test whether 3 additional factors–living below the poverty level, living in a pre-1986 home, and living near a known polluting source–might predict child BLLs, controlling for sex and age. For the models using the imputed dataset, the interaction of sex and age was consistently significant; only living near a known polluting source predicted child BLLs and the significant increase was small.

The variability of BLLs observed in our study support the strong influence on variability that might be expected from multiple shifting environmental, physiological, and behavioral factors, that together in complex ways, influence Pb exposure, absorption, distribution, and accumulation in children [22]. A critical aspect of current policy is the implicit assumption that only children below the ages of either 3 years (in a majority of states) or 6 years (in many fewer states) are at a level of Pb exposure risk to warrant testing. While behaviors such as crawling, teething, mouthing contaminated non-food items, and playing in soil, place younger children at greater risk of Pb exposure and absorption, there is no reason to suggest that older children are at so little risk that testing should not be required. Indeed, the 2012 CDC Lead Prevention Task Force recognized the lack of evidence for restricting screening for children beyond the age of 6 years, and discussed the critical importance of identifying behaviors and factors that may predict BLLs in older children, and in children with chronic low-level Pb absorption [43].

Previous studies examining the stability of BLLs over time in children with chronic low-level Pb absorption have rarely been conducted, and those that have been done, have focused only on infants. For example, in a very large study ($N$ = 267,687) of children ages 1 to 3 years, BLL changes among children with BLLs < 10 μg/dL were studied over a 36-month period. Conducted in the 1990s, change in the BLLs of children screened within two weeks of their first or second birthdays, was predicted by time, age, and season [44]. The findings were difficult to interpret due to the overlap of the study with the phasing out of Pb paint, leaded gasoline, and of the residential use of Pb solder and Pb pipes. Perhaps not surprisingly, among these youngest children, BLLs showed a steady decrease.

Another perhaps less confounded but also very short-term (4 month) study analyzed BLLs from a relatively small sample of 42 children during the first two years of life and measured Pb concentrations and isotope ratios in blood, urine, household dust, and duplicate diet samples (bottle-fed vs. breastfed) for children with BLLs ≤ 10 μg/dL [45]. The geometric mean of BLLs from birth and at 4 months remained unchanged (0.92 and 0.99 μg/dL, respectively). After 4 months, the BLLs remained near-constant or slightly increased, depending on whether children were exposed to Pb during home renovations for one brief or prolonged exposure, and whether bottle or breastfed.

Neither of these studies addressed change in BLLs for children across childhood, and neither examined dangerous lowest-range BLLs that are most common today. Many more studies are needed to characterize BLL change over time (for example, over a minimum of 5 years with BLL testing at least every 4 months), in order to identify sources that explain chronic low-level Pb, and to understand how Pb is absorbed and distributed at different developmental ages.

While many more studies are needed to better understand how BLLs represent child Pb exposure across the childhood years, overall, these findings provide initial support for the notion that our current child BLL testing policies are in need of major revision. One or two BLL tests for only youngest children are inadequate for detecting shifts in Pb exposure that, based on the data presented here, appear to be common among children living in high-risk neighborhoods throughout the school-age years. Previously our group presented a feasible model and plan for bi-annual universal testing, which shows how current testing practices could be amended to accomplish bi-annual testing in high-risk neighborhoods using (capillary) finger-stick samples collected following strict methods for ensuring clean sample collection and analyzed by ICPMS or GFAAS with a limit of detection at $\leq 0.04$ μg/dL [46].

An alternative to re-instating bi-annual universal testing for all children may be to identify through known risk factor modeling, groups of children who are likely to be currently exposed to Pb. A substantial amount of work in this area is needed. Current prediction models of child Pb exposure risk often include a combination of sociodemographic and housing characteristics, distance from point sources of Pb, and family behaviors to determine exposure risk in early childhood [47, 48]. They do not however adequately capture relatively complete combinations of environmental and endogenous Pb sources, nor their variability, given age-specific behaviors that may influence children's exposures. It is perhaps not surprising that these models have proven to have low predictive power. Some progress has been made in the use of data from the National Health Administration and Nutritional Examination Survey (NHANES) for the prediction of state and local child BLLs, but thus far, even these models have succeeded in predicting only (approximately) 30% of exposures [49].

Given the importance of predicting child lead exposure with greater accuracy, future studies need to continue characterizing factors that explain lower range BLLs in high-risk children. Studies should consider other key social, economic, and environmental factors that are not often considered in risk prediction models or for explaining the variability of children's Pb exposure during childhood. These could include for example factors such as food insecurity, housing instability, and exposure to tobacco products [50].

## Limitations

Data collection for this study was interrupted by COVID shutdowns. During the initial weeks of shutdown, we expected that data collection would resume within one or two months, if not weeks. We shifted our activities to virtual venues in the hope of promoting family engagement and maintained monthly or bi-monthly contact with all families via telephone. As the pandemic wore on, we developed educational videos on Pb prevention that were posted on our study website. By the time it was safe to return to the community, our study team had mostly graduated and moved on to other positions, and the study could not be resumed. As a result, multiple imputation had to be used to allow for regression analyses and, while not ideal, the close similarity of results obtained with the original as compared to imputed data sets gave some confidence that the results from the imputed dataset were plausible (Table 5). Nonetheless, we interpret these results with caution. A less impactful limitation concerned the pre-pandemic challenges of completing BLL tests every 3 months with families in which nearly all of the parents worked full time. While the statistical models used readily accommodated unequal timing between test waves, a dataset with more equal time intervals would allow for a greater range of hypotheses regarding child BLL change over time.

## Summary

This study suggested that BLL variability is substantial over time, and did not diminish with age, in children ages 6 months to 16 years, and living in neighborhoods at high-risk of Pb

exposure. The findings further suggested that in high-risk neighborhoods, child Pb sources posed shifting risks for children, and thus children require ongoing BLL monitoring throughout the school age years. Given the amount of variability observed, bi-annual routine BLL testing should be a minimum goal, to ensure detection of shifting lead exposure for children living in neighborhoods at high risk of Pb exposure, and/or for those living near known industrial sources of Pb emissions and/or interstate highways. A financially and practically feasible approach for re-instituting universal testing has been described in detail [46].

## Supporting information

**S1 File. BLLs stability database.**
(SAV)

**S2 File. SPSS syntax for models.**
(DOCX)

## Acknowledgments

The authors would like to acknowledge Alexander Obeng; Diana Moreno; Crystal Costa; Carlos Chavarria; and Jaleen Avila for their important contributions to recruitment for this study.

## Author Contributions

**Conceptualization:** Michelle Del Rio, Ganga M. Hettiarachchi, Christina Sobin.

**Data curation:** Michelle Del Rio, Christina Rodriguez, Elizabeth Alvarado Navarro, Ganga M. Hettiarachchi, Christina Sobin.

**Formal analysis:** Michelle Del Rio, Chandima Wekumbura, Madhubhashini B. Galkaduwa, Ganga M. Hettiarachchi, Christina Sobin.

**Funding acquisition:** Michelle Del Rio, Ganga M. Hettiarachchi, Christina Sobin.

**Investigation:** Michelle Del Rio, Christina Rodriguez, Elizabeth Alvarado Navarro, Chandima Wekumbura, Madhubhashini B. Galkaduwa, Ganga M. Hettiarachchi, Christina Sobin.

**Methodology:** Michelle Del Rio, Ganga M. Hettiarachchi, Christina Sobin.

**Project administration:** Michelle Del Rio, Christina Sobin.

**Resources:** Michelle Del Rio, Christina Rodriguez, Elizabeth Alvarado Navarro, Chandima Wekumbura, Madhubhashini B. Galkaduwa, Ganga M. Hettiarachchi, Christina Sobin.

**Supervision:** Michelle Del Rio, Christina Sobin.

**Validation:** Michelle Del Rio, Christina Rodriguez, Elizabeth Alvarado Navarro, Chandima Wekumbura, Madhubhashini B. Galkaduwa, Ganga M. Hettiarachchi, Christina Sobin.

**Visualization:** Christina Sobin.

**Writing – original draft:** Michelle Del Rio, Christina Sobin.

**Writing – review & editing:** Michelle Del Rio, Christina Rodriguez, Elizabeth Alvarado Navarro, Chandima Wekumbura, Madhubhashini B. Galkaduwa, Ganga M. Hettiarachchi, Christina Sobin.

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
