## [Decision Letter · Decision Letter 0]

20 Apr 2023

PONE-D-22-34079Stability of blood lead levels in children with low-level lead absorptionPLOS ONE

Dear Dr. Michelle

Thank you for submitting your manuscript to PLOS ONE. After careful consideration, we feel that it has merit but does not fully meet PLOS ONE’s publication criteria as it currently stands. Therefore, we invite you to submit a revised version of the manuscript that addresses the points raised during the review process.

Please submit your revised manuscript within Jun 04 2023 11:59PM. If you will need more time than this to complete your revisions, please reply to this message or contact the journal office at plosone@plos.org. Please include the following items when submitting your revised manuscript:A rebuttal letter that responds to each point raised by the academic editor and reviewer(s). You should upload this letter as a separate file labeled 'Response to Reviewers'.A marked-up copy of your manuscript that highlights changes made to the original version. You should upload this as a separate file labeled 'Revised Manuscript with Track Changes'.An unmarked version of your revised paper without tracked changes. You should upload this as a separate file labeled 'Manuscript'.If applicable, we recommend that you deposit your laboratory protocols in protocols.io to enhance the reproducibility of your results. Protocols.io assigns your protocol its own identifier (DOI) so that it can be cited independently in the future. For instructions see: https://journals.plos.org/plosone/s/submission-guidelines#loc-laboratory-protocols. Additionally, PLOS ONE offers an option for publishing peer-reviewed Lab Protocol articles, which describe protocols hosted on protocols.io. Read more information on sharing protocols at https://plos.org/protocols?utm_medium=editorial-email&utm_source=authorletters&utm_campaign=protocols.

We look forward to receiving your revised manuscript.

Kind regards,

Shailja Sharma, MD Biochemistry

Academic Editor

PLOS ONE

Journal Requirements:

Dear Dr Michelle

Please find details of the reviewers comments.

Reviewer 1 Comments

Very well written, and I appreciate this important work. This is a very important work for the sectors. I like the way the findings are presented. However, a few minor comments might improve the article” Stability of blood lead levels in children with low-level lead absorption”

1. It would be great to inform sample transportation time from collection to lab (line 157). In addition, having a very brief of data collectors and their qualifications.

2. Adding a sentence on policy implications would enrich the article and may be included in the conclusion of the discussion section.

3. As the study didn’t look at the impact of interventions discussed in the manuscript, do they want to recommend any future research?

Reviewer 2 Comments:

In the manuscript entitled "Stability of blood lead levels in children with low-level lead absorption" the authors suggest minimum biannual BLL monitoring throughout childhood, particularly for those living in under-served neighborhoods at high-risk of Pb exposure, and for those living near active industrial sites or major sources of vehicle exhaust pollution.

Manuscript is overall well written.

Some minor changes are as follows:

Abstract is too big and may be shortened

Paper needs a proper proofreading as there are certain grammatical and syntax errors.

Reviewers' comments:

Reviewer's Responses to Questions

**Comments to the Author**

1. Is the manuscript technically sound, and do the data support the conclusions?

Reviewer #1: Yes

Reviewer #2: Yes

2. Has the statistical analysis been performed appropriately and rigorously? 

Reviewer #1: Yes

Reviewer #2: Yes

3. Have the authors made all data underlying the findings in their manuscript fully available?

Reviewer #1: Yes

Reviewer #2: Yes

4. Is the manuscript presented in an intelligible fashion and written in standard English?

Reviewer #1: Yes

Reviewer #2: Yes

5. Review Comments to the Author

Reviewer #1: This is an interesting an important paper and has future implications to the lead work globally. However, as the findings supported by small sample size, might recommend a future research with a larger sample size . However, I am happy to accept the manuscript for publication.

Reviewer #2: In the manuscript entitled "Stability of blood lead levels in children with low-level lead absorption" the authors suggest minimum biannual BLL monitoring throughout childhood, particularly for those living in under-served neighborhoods at high-risk of Pb exposure, and for those living near active industrial sites or major sources of vehicle exhaust pollution.

Manuscript is overall well written.

Some minor changes are as follows:

Abstract is too big and may be shortened

Paper needs a proper proofreading as there are certain grammatical and syntax errors.

6. PLOS authors have the option to publish the peer review history of their article (what does this mean?). If published, this will include your full peer review and any attached files.

Reviewer #1: **Yes: **Dr. Md. Mahbubur Rahman

Reviewer #2: No

---

## [Author Response · Author response to Decision Letter 0]

29 May 2023

Thank you for forwarding the reviewers’ comments for our manuscript entitled “Stability of blood lead levels in children with low-level lead absorption” submitted for re-view to PLOS ONE. We have incorporated all suggestions by both reviewers. We believe these revisions have resulted in an improved manuscript.

---

## [Editor Report · Decision Letter 1]

5 Jun 2023

Stability of blood lead levels in children with low-level lead absorption

PONE-D-22-34079R1

Dear Michelle Del Rio

We’re pleased to inform you that your manuscript has been judged scientifically suitable for publication and will be formally accepted for publication once it meets all outstanding technical requirements.

Kind regards,

Shailja Sharma, MD Biochemistry

Academic Editor

PLOS ONE

---

## [Editor Report · Acceptance letter]

16 Jun 2023

PONE-D-22-34079R1 

Stability of blood lead levels in children with low-level lead absorption 

Dear Dr. Del Rio:

I'm pleased to inform you that your manuscript has been deemed suitable for publication in PLOS ONE. Congratulations! Your manuscript is now with our production department. 

Kind regards, 

on behalf of

Dr. Shailja Sharma 

Academic Editor

PLOS ONE